# Health literacy and preventive behaviors toward microplastic contamination among communities in the major river basins of northeastern Thailand

Santisith Khiewkhern[1,2]*, Supatra Noo-In[1], Chitkamon Srichomphoo[3], Jirarat Ruetrakul [4], Ruchakron Kongmant [2], Nitikorn Phoosuwan [5,6]*

1 Faculty of Public Health, Mahasarakham University, Mahasarakham, Thailand, 2 Public Health and Environmental Policy in Southeast Asia Research Cluster (PHEP-SEA), Mahasarakham, Thailand, 3 Faculty of Science and Technology, Nakhon Pathom Rajabhat University, Nakhon Pathom, Thailand, 4 Faculty of Nursing, Naresuan University, Phitsanulok, Thailand, 5 Department of Public Health and Caring Sciences, Uppsala University, Uppsala, Sweden, 6 Faculty of Public Health, Thammasat University, Pathum Thani, Thailand

* nitikorn.phoosuwan@uu.se, nitikorn.p@fph.tu.ac.th (NP); santisith.k@msu.ac.th (SK)

## Abstract

### Background

Microplastic contamination in freshwater systems poses increasing environmental and public health risks, particularly for riverine communities. Health literacy may critically influence individuals' capacity to recognize exposure pathways and adopt preventive behaviors.

### Objective

This study assessed health literacy and preventive behaviors related to microplastic contamination and examined factors associated with poor health literacy and preventive practices among communities in northeastern Thailand.

### Methods

A cross-sectional survey was conducted among 942 adults residing along the Mekong, Chi, and Mun River basins. Health literacy and preventive behaviors were assessed using a structured questionnaire. Multivariable logistic regression was performed to identify associated factors, whilst adjusted odds ratios (aOR) with 95% confidence intervals (CI) and statistical significance set at $p < 0.05$.

### Results

Nearly half of the participants exhibited insufficient health literacy, with decision-making identified as the weakest domain. While approximately half reported good preventive behaviors, insufficient health literacy was the strongest determinant of poor preventive practices (aOR=9.18; 95%CI: 6.62–12.72). Residents of rural

**Data availability statement:** All XXX files are available from the https://osf.io/46txe/overview.

**Funding:** The manuscript has not been published or submitted for publication elsewhere. The authors received no specific funding for this work. The funder (Mahasarakham University) had no role in study design, data collection and analysis, decision to publish, or preparation of the manuscript. All authors have approved the content of the manuscript and have contributed significantly to the research involved/the writing of the manuscript.

**Competing interests:** The authors have declared that no competing interests exist.

(aOR=6.47; 95%CI: 4.47–9.37) and semi-urban (aOR=2.45; 95%CI: 1.76–3.41) communities and younger age groups were significantly more likely to have poor health literacy. Community type (semi-urban: aOR=2.61; 95%CI: 1.81–3.78) and longer duration of residence (>60 years: aOR=2.36; 95%CI: 1.37–4.06) were also significantly associated with preventive behaviors.

## Conclusion

Health literacy is a critical determinant of microplastic-related preventive behaviors. Interventions targeting higher-order literacy skills, particularly decision-making, should be integrated into environmental health and public health strategies to reduce microplastic exposure and promote sustainable preventive practices in riverine communities.

## Introduction

Plastics have become an essential part of modern life due to their durability, versatility, and affordability. However, the overproduction and mismanagement of plastic waste have caused severe environmental pollution, particularly through the generation of microplastics (MPs)—plastic fragments smaller than 5 mm—that persist in aquatic and terrestrial ecosystems [1,2]. Since the 1950s, more than 8.3 billion metric tons of plastics have been produced, of which only 9% have been recycled, while the rest accumulate in landfills and the natural environment [3]. These plastic materials gradually fragment under environmental stressors such as UV radiation, abrasion, and microbial activity, resulting in persistent microplastics that threaten ecosystems worldwide [4,5].

Microplastics are now recognized as an emerging global pollutant due to their wide distribution, persistence, and potential toxicity [6,7]. They enter the environment from multiple sources, including wastewater discharge, industrial effluents, stormwater runoff, and the breakdown of larger plastic debris [8,9]. Once released, they can be transported through waterways and accumulate in sediments and aquatic organisms [10] Studies across Asia have demonstrated widespread microplastic contamination, particularly in low- and middle-income countries where waste management systems remain inadequate and enforcement mechanisms are limited [11,12]. For example, Pinto et al. [13] and Alam et al. [14] found that rivers in Thailand and Indonesia near industrial and residential areas contain high concentrations of microplastics due to mismanaged waste and insufficient treatment infrastructure. These findings suggest that rivers act as major transport pathways of microplastics from land-based sources to marine environments [15].

Microplastics are not only an environmental hazard but also a public health concern. They have been detected in marine and freshwater organisms, including plankton and fish, indicating potential trophic transfer across aquatic food webs and raising concerns about bioaccumulation in higher trophic levels [16,17]. Laboratory studies show that exposure to microplastics can lead to oxidative stress, inflammation, and

reproductive toxicity in aquatic species [18]. Moreover, microplastics can absorb and carry toxic chemicals such as heavy metals and persistent organic pollutants, enhancing their bioavailability and health risks to both wildlife and humans [19,20]. Human exposure primarily occurs through ingestion of contaminated food and drinking water, inhalation of airborne particles, and possible dermal contact [20,21]. Recent biomonitoring studies have detected microplastics in human stool samples and placental tissues, raising concerns about potential systemic exposure and long-term biological implications [22,23].

Technological interventions have been developed to address microplastic pollution. Conventional wastewater treatment can remove a portion of microplastics through sedimentation and filtration, while coagulation–flocculation and ultrafiltration improve efficiency [24,25]. Advanced treatments using polyaluminum chloride, ferric chloride, and chitosan-based coagulants have demonstrated enhanced removal efficiency, particularly for microplastics smaller than 10 μm [26–28]. However, complete removal remains challenging, especially in rural communities that rely on untreated water sources [29]. In Thailand, studies have revealed microplastics in treated and untreated water supplies, as well as in aquatic organisms consumed by local populations [30–32]. Therefore, improving community awareness and behavior regarding microplastic exposure is crucial to complement technological solutions.

Health literacy is a key determinant influencing how individuals perceive and respond to environmental risks. It reflects people's ability to obtain, understand, evaluate, and apply information to make informed health-related decisions [31]. In the context of microplastic pollution, individuals with higher health literacy are more likely to engage in preventive behaviors such as reducing single-use plastics, choosing safe drinking sources, and supporting proper waste management [33].

Given the rising microplastic contamination in freshwater and human exposure risks, this study aimed to assess the health literacy and preventive behaviors among communities living in the major river basins of Northeastern Thailand. Understanding the factors associated with these behaviors can guide the development of effective community-based interventions, public awareness campaigns, and environmental health policies to reduce exposure and promote sustainable water and waste management practices.

## Methods

### Study design

This study employed a cross-sectional descriptive design to assess the levels of health literacy and preventive behaviors related to microplastic contamination among residents living along the three major river basins in Northeastern Thailand—the Chi, Mun, and Mekong Rivers. The study aimed to explore associations between demographic factors, health literacy dimensions, and preventive behaviors to identify determinants influencing microplastic exposure prevention in local communities.

### Study population

The target population consisted of adults aged 18 years and above residing in communities located within five kilometers of the main riverbanks in selected provinces representing the three river basins—Nakhon Phanom (Mekong River), Maha Sarakham (Chi River), and Buriram (Mun River). These provinces were chosen to represent different environmental and socioeconomic contexts, including agricultural, urban, and mixed-use areas.

### Samples and sample size

The sample size was calculated using a single-population proportion formula to achieve adequate statistical precision in estimating the prevalence of environmental health literacy in the target population. A 95% confidence level and an assumed proportion of 0.50 were applied, the latter chosen to maximize variability and provide the most conservative estimate in the absence of reliable prior data. Based on these parameters and a margin of error of approximately ±3.2%, the

minimum required sample size was calculated to be 942 participants. This sample size was considered sufficient to yield stable and reliable estimates for subsequent analyses and was therefore adopted for the study. This sample size ensured sufficient representation from each province. Inclusion criteria were adults who: (1) aged 18 years and above, (2) resided in the selected river basin communities, including temporary residents and migrants living near the border, (3) were able to read and understand Thai language, and (4) provided informed consent to participate in the study. Exclusion criteria included: (1) individuals with severe cognitive impairment or communication difficulties that prevented questionnaire completion, (2) temporary residents or migrants who had lived in the area for less than 6 months, and (3) participants who did not complete more than 80% of the questionnaire.

## Sampling method

A multistage sampling technique was applied in this study. **Stage 1** involved the purposive selection of three provinces to represent the distinct community types based on their geographical coverage, administrative boundaries, population density, and location within the river basins. The provinces selected were Nakhon Phanom (representing Urban communities), Maha Sarakham (representing Semi-urban communities), and Buriram (representing Rural communities), **Stage 2:** From each province, two districts located near the riverbanks were randomly selected, **Stage 3:** Within each district, two communities were systematically selected based on population size and accessibility, and **Stage 4:** Households were randomly selected from community lists, and one eligible adult per household was chosen using a simple random method. This approach ensured representation across various environmental and socioeconomic backgrounds, increasing the generalizability of the findings.

## Ethical considerations

The study protocol was reviewed and approved by the Human Research Ethics Committee of Maha Sarakham University (No. 556–524/2567 dated 29 August 2024–28 August 2025)**.** All participants provided informed consent prior to participation, and their confidentiality and anonymity were strictly maintained throughout the study.

## Data collection procedures

Data were collected from 15 January to 31 March 2025 by trained research assistants through face-to-face interviews. Prior to data collection, enumerators underwent a one-day training session to standardize data collection procedures and ensure ethical compliance. Each participant received an explanation of the study purpose, potential benefits, and confidentiality measures before providing written informed consent. Data were recorded anonymously using coded questionnaires. Quality control was maintained by daily verification of completed forms and random re-checks of 10% responses by field supervisors.

## Outcome measures

The study used a structured questionnaire developed based on existing literature and the Nutbeam Health Literacy Framework [31]. It comprised three sections:

Demographic information (e.g., age, gender, education, occupation, income, and water source).

Health literacy assessment: Six components—cognitive understanding, access to information and services, communication, self-management (action), appraisal, and decision-making—were measured using a five-point Likert scale, with higher scores indicating better literacy. The total health literacy score was interpreted using an 80% threshold, whereby scores below 80% indicated insufficient health literacy, while scores of 80% or higher reflected a sufficient level of health literacy [34].

Preventive behavior assessment: Items measured frequency and consistency of microplastic contamination prevention actions, including safe water consumption, waste segregation, and reduced plastic use. The total preventive behavior

 

score was interpreted using an 80% threshold, whereby scores below 80% indicated poor preventive behavior, while scores of 80% or higher reflected a good level of preventive behavior.

Reliability testing using Cronbach's alpha yielded coefficients of 0.89 for the overall scale and ranged between 0.81–0.87 for subcomponents, indicating good internal consistency. This instrument was developed based on established health literacy frameworks and previous environmental health literacy studies. However, this study represents an initial application of the microplastic health literacy scale, and comprehensive construct validation, including factor analysis, was not conducted.

### Data analysis

Data were analyzed using IBM SPSS Statistics version 26 (IBM Corp., Armonk, NY, USA). *Descriptive statistics*—frequency, percentage, mean, and standard deviation—were used to describe participant characteristics, health literacy scores, and preventive behaviors.

*Inferential statistics* were applied as follows: (1) Independent *t*-test to compare mean behavior scores across demographic variables, (2) Chi-square test and Fisher exact test to examine associations between categorical variables and levels of preventive behavior, (3) Binary logistic regression was used to assess the relationships between demographic variables and health literacy and preventive behaviors. Variables with a p-value ≤ 0.25 in the univariable analysis were included in the multivariable analysis. The backward selection method was applied for the multivariable logistic regression. Model fit was evaluated using the Hosmer–Lemeshow goodness-of-fit test. Crude Odds Ratio (Crude OR) and adjusted OR (adj.OR) were calculated. Statistical significance was set at $p < 0.05$. All analyses followed ethical guidelines and research integrity standards.

Occupation, household size, and water source were included as contextual socioeconomic and environmental variables and screened in univariable analyses prior to multivariable modeling. Detailed occupational exposure assessment was not conducted, as the study focused on community-level health literacy and preventive behaviors rather than task-specific exposure pathways.

## Results

The analysis of 942 participants found a near-equal distribution of health behaviors, with 463 (49.20%) categorized as having "Good behavior" and 479 (50.80%) as having "Poor behavior". Regarding community type, the "Good behavior" group was predominantly from urban communities (228, 49.20%), while the "Poor behavior" group was most frequently found in semi-urban communities (209, 43.60%). For gender, females constituted the majority in both groups, whilst the age distribution showed that the "Good behavior" group had a mean age of 50.56 ± 14.48 years, and the "Poor behavior" group had a mean age of 48.39 ± 16.22 years. In terms of Occupation, Agriculture/Fishery was the largest group for both bahvior groups, whereas the "Good behavior" group had a mean of 7,333.29 ± 5,959.00 baht and the "Poor behavior" group had a higher mean of 9,055.34 ± 7,975.66 baht. See Table 1.

### Health literacy level with six components

The mean scores and standard deviations for the six components of health literacy related to microplastics contamination prevention among the 942 participants. The analysis revealed that the overall health literacy level was good, with a mean score of 2.34 ± 0.46. Examining the specific components, four components were rated at a good level: Communication (2.49 ± 0.56), Action (2.40 ± 0.51), Appraisal (2.38 ± 0.58), and Access to Health Information and Service (2.34 ± 0.55). Conversely, one component was rated as Moderate: Cognitive (or Understanding), with a mean score of 2.30 ± 0.52. Most notably, the component of Decision-Making scored the lowest (mean = 1.61 ± 0.47) and was the only component rated as Poor. See Table 2.

**Table 1. General information of participants classified by behavior in preventing microplastic contamination (n = 942).**

| Factors | Behaviors | | | | Total (n = 942) | | p-value |
|---|---|---|---|---|---|---|---|
| | Good (n = 463) | | Poor (n = 479) | | | | Exact test/*t*-test |
| | n | (%) | n | (%) | n | (%) | |
| **Communities** | | | | | | | 0.001** |
| Urban communities | 228 | (49.20) | 111 | (23.20) | 339 | (36.00) | |
| Semi-urban communities | 127 | (27.40) | 209 | (43.60) | 336 | (35.70) | |
| Rural communities | 108 | (23.30) | 159 | (33.20) | 267 | (28.30) | |
| **Gender** | | | | | | | 0.409 |
| Male | 162 | (35.00) | 180 | (37.60) | 342 | (36.30) | |
| Female | 301 | (65.00) | 299 | (62.40) | 600 | (63.70) | |
| **Age (years)** | | | | | | | |
| Mean (SD) | 50.56 | (14.48) | 48.39 | (16.22) | 49.46 | (15.42) | 0.031* |
| 18–45 years | 162 | 35.00 | 205 | 42.80 | 367 | 39.00 | 0.024* |
| 46–60 years | 182 | 39.30 | 152 | 31.70 | 334 | 35.50 | |
| >60 years | 119 | 25.70 | 122 | 25.50 | 241 | 25.60 | |
| **Occupation** | | | | | | | |
| Agriculture/ Fishery | 178 | (38.40) | 175 | (36.50) | 353 | (37.50) | 0.855 |
| Trade | 86 | (18.60) | 85 | (17.70) | 171 | (18.20) | |
| Labor | 113 | (24.40) | 127 | (26.50) | 240 | (25.50) | |
| Government/State enterprise/Others | 86 | (18.60) | 92 | (19.20) | 178 | (18.90) | |
| **Income per month (baht)** | | | | | | | |
| Mean (SD) | 7333.29 | (5959.65) | 9055.34 | (7975.66) | 8,208.92 | (7,105.80) | < 0.001** |
| ≤ 10,000 baht | 318 | (68.70) | 283 | (59.10) | 601 | (63.80) | 0.002** |
| > 10,000 baht | 145 | (31.30) | 196 | (40.90) | 341 | (36.20) | |
| **Duration of residence** (years) | | | | | | | |
| Mean (SD) | 43.8 | (18.54) | 42.80 | (18.93) | 43.30 | (18.74) | 0.416 |
| **Duration of residence (years)** | 31 | 6.70 | 33 | 6.90 | 64 | 6.80 | 0.003** |
| ≤ 20 years | 72 | 15.50 | 67 | 14.00 | 139 | 14.80 | 0.083 |
| 21–40 years | 118 | 25.50 | 151 | 31.50 | 269 | 28.60 | |
| 41–60 years | 191 | 41.30 | 166 | 34.70 | 357 | 37.80 | |
| >60 years | 82 | 17.70 | 95 | 19.80 | 177 | 18.80 | |
| **Religion** | | | | | | | 0.587 |
| Buddhist | 451 | (97.40) | 465 | (97.10) | 915 | (97.20) | |
| Others | 12 | (2.60) | 14 | (2.90) | 26 | (2.80) | |
| **Education** | | | | | | | 0.001** |
| No formal education | 12 | (2.60) | 8 | (1.70) | 20 | (2.10) | |
| Primary | 256 | (55.30) | 197 | (41.10) | 453 | (48.10) | |
| Secondary | 128 | (27.60) | 140 | (29.20) | 268 | (28.50) | |
| Vocational | 35 | (7.60) | 47 | (9.80) | 82 | (8.70) | |
| Bachelor's or higher | 32 | (6.90) | 87 | (18.20) | 119 | (12.60) | |
| **Family size** | | | | | | | |
| Small (1–2 persons) | 68 | (14.70) | 97 | (20.30) | 165 | (17.50) | 0.075 |
| Medium (3–4 persons) | 218 | (47.10) | 216 | (45.10) | 434 | (46.10) | |
| Large (≥5 persons) | 177 | (38.20) | 166 | (34.70) | 343 | (36.40) | |
| **Drinking water source** | | | | | | | 0.030* |
| River | 7 | (1.50) | 8 | (1.70) | 15 | (1.60) | |
| Tap water | 36 | (7.80) | 45 | (9.40) | 81 | (8.60) | |

*(Continued)*

**Table 1.** (Continued)

| Factors | Behaviors | | | | Total (n = 942) | | p-value |
|---|---|---|---|---|---|---|---|
| | Good (n = 463) | | Poor (n = 479) | | | | Exact test/*t*-test |
| | n | (%) | n | (%) | n | (%) | |
| Filtered water | 142 | (30.70) | 118 | (24.60) | 260 | (27.60) | |
| Bottled water | 256 | (55.30) | 298 | (62.20) | 554 | (58.80) | |
| Others | 22 | (4.80) | 10 | (2.10) | 32 | (3.40) | |
| **Household water source** | | | | | | | < 0.001** |
| River | 41 | (8.90) | 26 | (5.40) | 67 | (7.10) | |
| Tap water | 358 | (77.30) | 389 | (81.20) | 747 | (79.30) | |
| Filtered water | 32 | (6.90) | 32 | (6.70) | 64 | (6.80) | |
| Bottled water | 9 | (1.90) | 24 | (5.00) | 33 | (3.50) | |
| Others | 23 | (5.00) | 8 | (1.70) | 31 | (3.30) | |
| **Illness in the past month** | | | | | | | |
| Yes | 89 | (19.20) | 122 | (25.60) | 211 | (22.40) | 0.020* |
| No/unsure | 374 | (80.80) | 355 | (74.40) | 729 | (77.60) | |
| **Visiting a physician in the past month** | | | | | | | |
| Yes | 90 | (19.40) | 140 | (29.20) | 230 | (24.40) | 0.001** |
| No/unsure | 373 | (80.60) | 339 | (70.80) | 712 | (75.60) | |
| **Type of illness in the past month** | | | | | | | 0.002* |
| Respiratory system | 35 | (7.60) | 65 | (13.60) | 100 | (10.60) | |
| Digestive system | 36 | (7.80) | 38 | (7.90) | 74 | (7.90) | |
| Circulatory/neurological | 23 | (5.00) | 40 | (8.40) | 63 | (6.70) | |
| None | 369 | (79.70) | 336 | (70.10) | 705 | (74.80) | |
| **Aquatic animal consumption (past month)** | | | | | | | 0.249 |
| Yes | 430 | (92.90) | 435 | (90.80) | 865 | (91.80) | |
| No/Unsure | 33 | (7.10) | 44 | (9.20) | 77 | (8.20) | |
| **Source of aquatic animals consumed** | | | | | | | 0.306 |
| Self-caught | 101 | (21.80) | 89 | (18.60) | 190 | (20.20) | |
| Market | 327 | (70.60) | 334 | (71.80) | 671 | (71.20) | |
| From friends/relatives | 35 | (7.60) | 46 | (9.60) | 81 | (8.60) | |
| **Type of aquatic animal Consumed** | | | | | | | 0.102 |
| Fish | 411 | (88.80) | 408 | (85.20) | 819 | (86.90) | |
| Others | 52 | (11.20) | 71 | (14.80) | 123 | (13.10) | |

**p-value<0.01, *p-value<0.05, SD=Standard deviation, 1 US dollar=32.50 baht.

## Health literacy level

The results indicate a nearly balanced distribution between the two levels. Specifically, 50.20% (n = 473) of the participants were found to have "sufficient health literacy", while a large proportion, 49.80% (n = 469), were categorized as having "Insufficient health literacy". This finding suggested that, despite the importance of health literacy in addressing environmental health issues, nearly half of the study population lacks the adequate level of literacy required to fully obtain, understand, evaluate, and apply microplastic-related health information effectively. This highlighted a critical need for targeted interventions to boost comprehension and decision-making skills within the communities. Although the overall mean health literacy score fell within the good range, domain-specific classification revealed a high prevalence of insufficient

**Table 2. Health literacy level with 6 components for microplastics contamination (n = 942).**

| Health literacy | Mean | SD. | Level |
|---|---|---|---|
| Cognitive | 2.30 | 0.52 | Moderate |
| Access to health information and service | 2.34 | 0.55 | Good |
| Communicate | 2.49 | 0.56 | Good |
| Action | 2.40 | 0.51 | Good |
| Appraisal | 2.38 | 0.58 | Good |
| Decision | 1.61 | 0.47 | Poor |
| Overall | 2.34 | 0.46 | Good |

SD. = Standard deviation.

health literacy, driven primarily by lower scores in higher-order domains such as critical appraisal and decision-making. See Table 3.

## Health behavior

The analysis revealed a near-even split in preventive practices, with the majority of participants categorized as having Poor Health Behaviors, accounting for 50.80% (n = 479) of the total sample. Conversely, those demonstrating "Good preventive behaviors" represented 49.20% (n = 463). This outcome, while showing a slight majority toward insufficient practices, underscores that nearly half of the community engages in positive actions to mitigate microplastic exposure. Nonetheless, the substantial proportion of the population still exhibiting poor health behaviors highlights the urgent requirement for targeted public health strategies and continued educational efforts to enhance consistent and sustainable preventive behaviors across the community. See Table 4.

## Factors associated with poor health literacy

The multivariable logistic regression model examined the factors associated with poor health literacy for microplastics contamination prevention. The model demonstrated an acceptable fit (Hosmer and Lemeshow Test p = 0.104) and accounted for 19.60% of the variance in poor health literacy (Nagelkerke $R^2$ = 0.196), correct classifying 66.00% of the observations.

The analysis identified several factors significantly associated with poor health literacy, using Urban community, Age > 60 years, ≤ 20 years duration of residence, and no previous illness as the reference groups (Adj.OR = 1). The

**Table 3. Health literacy level for Preventing microplastics contamination (n = 942).**

| Health literacy | Number (Person) | Percentage |
|---|---|---|
| Sufficiency | 473 | 50.20 |
| Insufficiency | 469 | 49.80 |
| Total | 942 | 100.00 |

**Table 4. Health behavior level for preventing microplastics contamination (n = 942).**

| Health behavior | Number (Person) | Percentage |
|---|---|---|
| Good | 463 | 49.20 |
| Poor | 479 | 50.80 |
| Total | 942 | 100.00 |

 

strongest independent predictor was Community type. Individuals residing in a rural community were 6.47 times more likely to have poor health literacy compared to those in an urban community (Adj.OR = 6.47, 95% CI: 4.47 to 9.37, p < 0.001). Similarly, those in a Semi-urban community were also significantly more likely to have Poor Health Literacy (Adj.OR = 2.45, 95% CI: 1.76 to 3.41, p < 0.001).

Age was also a significant factor. Compared to those aged > 60 years, individuals in the 18–45 years group were 3.53 times more likely to have Poor Health Literacy (Adj.OR = 3.53, 95% CI: 1.90 to 6.55, p < 0.001), and those in the 46–60 years group were 2.35 times more likely (Adj.OR = 2.35, 95% CI: 1.28 to 4.34, p = 0.006).

Regarding Duration of residence, only the category of > 60 years was significantly associated, showing a protective effect where these individuals were less likely to have Poor Health Literacy (Adj.OR = 0.49, 95% CI: 0.25 to 0.96, p = 0.038) compared to the ≤ 20 years group. Finally, certain History of illness groups showed a higher likelihood of Poor Health Literacy compared to having no previous illness[9]. Specifically, those with a history of respiratory diseases were 1.64 times more likely to have poor health literacy (Adj.OR = 1.64, 95% CI: 1.02 to 2.62, p = 0.039)[10], and those with a history of cardiovascular diseases were 1.90 times more likely (Adj.OR = 1.90, 95% CI: 1.07 to 3.38, p = 0.028). See Table 5.

## Factors associated with poor health behaviors

The multivariable logistic regression model successfully assessed the factors associated with poor health behaviors for preventing microplastics contamination, showing a good fit to the data (Hosmer and Lemeshow Test p = 0.961) and accounting for a substantial portion of the variance (Nagelkerke $R^2$ = 0.351). The overall percentage of correctly classified

**Table 5. Factor associated with poor health literacy for microplastics contamination prevention.**

| Factors | Health literacy | | Bivariable analysis | | Multivariable analysis | | |
|---|---|---|---|---|---|---|---|
| | Sufficiency | Insufficiency | Crude OR | *p*-value | Adj.OR | *p*-value | 95%CI of Adj.OR |
| | n (%) | n (%) | | | | | |
| **Communities** | | | | | | | |
| Urban | 241 (51.00) | 98 (20.90) | 1 | 0.001** | 1 | 0.001** | |
| Semi-urban | 154 (32.50) | 182 (38.80) | 2.91 | 0.001** | 2.45 | 0.001** | 1.76 to 3.41 |
| Rural | 78 (16.50) | 189 (40.30) | 5.96 | 0.001** | 6.47 | 0.001** | 4.47 to 9.37 |
| **Age (Years)** | | | | | | | |
| >60 years | 143(30.20) | 98 (20.90) | 1 | 0.001** | 1 | 0.001** | |
| 18–45 years | 152 (32.20) | 215 (45.80) | 2.06 | 0.001** | 3.53 | 0.001** | 1.90 to 6.55 |
| 46–60 year | 178 (37.60) | 156 (33.30) | 1.28 | 0.150 | 2.35 | 0.006* | 1.28 to 4.34 |
| **Duration of residence (years)** | | | | | | | |
| ≤20 years | 55 (11.60) | 84 (17.90) | 1 | 0.001** | 1 | 0.054 | |
| 21–40 years | 119 (25.20) | 150 (32.00) | 0.83 | 0.366 | 0.82 | 0.593 | 0.40 to 1.69 |
| 41–60 years | 195 (41.20) | 162 (34.50) | 0.54 | 0.003* | 0.57 | 0.100 | 0.29 to 1.11 |
| >60 years | 104 (22.00) | 73 (15.60) | 0.46 | 0.001** | 0.49 | 0.038* | 0.25 to 0.96 |
| **History of illness in the past month** | | | | | | | |
| No previous illness | 376(79.50) | 329 (70.20) | 1 | 0.007* | 1 | 0.020* | |
| Respiratory diseases | 37 (7.80) | 63 (13.40) | 0.51 | 0.003** | 1.64 | 0.039 | 1.02 to 2.62 |
| Gastrointestinal diseases | 31 (6.60) | 43 (9.20) | 0.63 | 0.063 | 1.60 | 0.084 | 0.94 to 2.72 |
| Cardiovascular diseases | 29 (6.10) | 34 (7.20) | 0.75 | 0.267 | 1.90 | 0.028* | 1.07 to 3.38 |

a. Variable(s) entered on step 1: Communities, Age group, Duration, Occupation, Family size, Income group, illness Type, Aquatic animals consumed.

b. Hosmer and Lemeshow Test, Chi-square = 13.234, p-value = 0.104; Nagelkerke R Square = 0.196; Overall Percentage = 66.00.

c. * *p*-value < 0.05; ** *p*-value < 0.01; Adj.OR = Adjusted Odds Raio, C.I. = Confidence interval.

observations was 75.20%. The analysis revealed that the most significant determinant of poor health behaviors was Health Literacy. Specifically, individuals categorized as having Insufficient Health Literacy were found to be 9.18 times more likely to exhibit poor prevention behaviors compared to those with Sufficient Health Literacy (Adj.OR = 9.18, 95% CI: 6.62 to 12.72, p < 0.001). Furthermore, Community Type played a significant role; those residing in a Semi-urban community were 2.61 times more likely to have poor behaviors than those in an Urban community (Adj.OR = 2.61, 95% CI: 1.81 to 3.78, p < 0.001). Lastly, Duration of residence of more than 60 years (>60 years) was associated with 2.36 times higher odds of poor health behaviors compared to those residing for 20 years or less (Adj.OR = 2.36, 95% CI: 1.37 to 4.06, p = 0.002). Other variables, such as family size and other duration categories, did not demonstrate a statistically significant association in the adjusted mode. See Table 6.

## Discussion

This study advances the current understanding of microplastic-related environmental health by elucidating the pivotal role of health literacy in shaping preventive behaviors among riverine communities in Northeastern Thailand. Although nearly half of the participants reported good preventive practices, a substantial proportion exhibited insufficient health literacy, particularly in higher-order domains such as cognitive understanding and decision-making. This discordance indicates that observed preventive behaviors may be largely habitual or externally reinforced rather than grounded in a comprehensive understanding of microplastic exposure pathways and associated health risks. Such a phenomenon reflects a broader challenge in environmental health communication, where behavioral compliance does not necessarily equate to informed agency [6,31].

**Table 6. Factor associated with poor health behaviors for preventing microplastics contamination.**

| Factors | Health behaviors | | Bivariable analysis | | Multivariable analysis | | |
|---|---|---|---|---|---|---|---|
| | Good | Poor | | | | | |
| | n (%) | n (%) | Crude OR | p-value | Adj.OR | p-value | 95%CI of Adj.OR |
| **Communities** | | | | | | | |
| Urban | 228 (49.30) | 111 (23.20) | 1 | 0.001** | 1 | 0.001** | |
| Semi-urban | 127 (27.40) | 209 (43.60) | 3.38 | 0.001** | 2.61 | 0.001** | 1.81 to 3.78 |
| Rural | 108 (23.30) | 159 (33.20) | 2.17 | 0.001** | 1.47 | 0.053 | 0.99 to 2.18 |
| **Duration of residence (years)** | | | | | | | |
| ≤20 years | 72 (15.50) | 67 (14.00) | 1 | 0.084 | 1 | 0.016* | |
| 21–40 years | 118 (25.50) | 151 (31.50) | 0.80 | 0.334 | 1.61 | 0.058 | 0.98 to 2.65 |
| 41–60 years | 191 (41.30) | 166 (34.70) | 1.11 | 0.609 | 1.42 | 0.148 | 0.88 to 2.65 |
| >60 years | 82 (17.70) | 95 (19.80) | 0.75 | 0.119 | 2.36 | 0.002** | 1.37 to 4.06 |
| **Family size** | | | | | | | |
| Large (≥5 persons) | 46 (9.90 | 60 (12.50) | 1 | 0.076 | | 0.064 | |
| Small (1–2 persons) | 268 (61.80) | 313 (65.30) | 1.52 | 0.029 | 1.30 | 0.261 | 0.82 to2.05 |
| Medium (3–4 persons) | 131 (28.30) | 106 (22.20) | 1.06 | 0.704 | 0.78 | 0.160 | 0.56 to 1.10 |
| **Health literacy** | | | | | | | |
| Sufficiency | 350 (75.60) | 35 (74.30) | 1 | 0.001** | | 0.001** | |
| Insufficiency | 113 (24.40) | 123 (25.70) | 8.96 | 0.001** | 9.18 | 0.001** | 6.62 to 12.72 |

a. Variable(s) entered on step 1: Communities, Age group, Duration, Occupation, Family size, Education, Family size, Income group, illness group, Visited Doctor, illness Type, Aquatic animals consumed, Health literacy group.

b. Hosmer and Lemeshow Test, Chi-square = 2.52, p-value = 0.961; Nagelkerke R Square = 0.351; Overall Percentage = 75.20.

c. * p-value < 0.05; ** p-value < 0.01; Adj.OR = Adjusted Odds Raio, C.I. = Confidence interval.

The markedly low score in the decision-making component warrants particular attention. According to Nutbeam's health literacy framework, decision-making represents a critical competency that integrates access, comprehension, appraisal, and action into coherent and sustained behavior [31]. The deficiency observed in this domain suggests that while participants may be capable of accessing and communicating information, they encounter difficulties in synthesizing evidence and translating it into rational, context-specific choices. Comparable patterns have been reported in studies of environmental and chemical risk perception, where individuals demonstrate surface-level awareness but limited capacity for critical evaluation and autonomous decision-making [33,35].

Pronounced spatial disparities in health literacy were evident, with residents of rural and semi-urban communities exhibiting substantially higher odds of insufficient health literacy than their urban counterparts. These findings underscore the influence of structural and contextual determinants, including inequities in access to environmental information, educational resources, and public health infrastructure. Previous studies in Southeast Asia have documented that river systems traversing agricultural and peri-urban landscapes often receive inadequately treated domestic and industrial effluents, resulting in elevated microplastic loads while simultaneously limiting opportunities for effective risk communication [13,14]. Consequently, communities facing the highest environmental exposure may paradoxically possess the lowest capacity to interpret and respond to such risks.

Age-related differences further illuminate the complex interplay between experiential knowledge and formal literacy. Younger participants demonstrated a higher likelihood of poor health literacy compared with older adults, a finding that diverges from conventional assumptions linking age to declining literacy. In this context, prolonged residence and cumulative exposure to environmental change may enhance contextual awareness and informal knowledge acquisition among older individuals, partially compensating for lower levels of formal education [36–38]. This highlights the importance of recognizing experiential learning as a complementary dimension of environmental health literacy.

Critically, this study demonstrates a robust and independent association between health literacy and preventive behaviors, with insufficient literacy emerging as the strongest predictor of poor preventive practices. In addition, the observed association between higher income and poorer preventive behaviors may reflect contextual urban–rural differences rather than economic capacity alone. Higher-income participants were more likely to reside in urbanized settings, where perceived reliance on infrastructural protection and lower perceived environmental exposure may reduce engagement in individual preventive practices. Conversely, lower-income residents in riverine communities may adopt more conservative behaviors, such as reliance on bottled water or traditional household practices, to mitigate perceived risks. These contextual factors were not explicitly modeled and should be interpreted cautiously. At first glance, the coexistence of a high prevalence of insufficient health literacy and a high level of good preventive behaviors may appear contradictory. However, this pattern reflects the multidimensional nature of health literacy. While many participants demonstrated adequate functional or interactive literacy that supports routine preventive practices, lower scores in higher-order domains—particularly critical appraisal and decision-making—resulted in overall classification as insufficient health literacy. In addition, preventive behaviors in riverine communities may be influenced by habitual practices, social norms, and contextual environmental factors that operate independently of comprehensive health literacy. Therefore, health literacy should be interpreted as a strong and independent predictor of preventive behaviors rather than the sole determinant.

In addition, the observed association between a history of respiratory illness and poorer health literacy should be interpreted with caution. Respiratory symptoms were self-reported and were not intended to indicate a direct or indirect effect of microplastic exposure. Given the cross-sectional design and the period of data collection, it is not possible to disentangle potential effects of microplastics from other environmental factors, particularly ambient air pollution and seasonal particulate matter exposure. Therefore, this finding reflects a contextual association rather than a causal relationship. This finding provides empirical support for theoretical models that position health literacy as a central mediator between environmental risk information and behavioral outcomes [37].

Individuals with limited literacy may struggle to evaluate the safety of drinking water sources, interpret information on microplastic contamination, or sustain plastic-reduction behaviors, thereby increasing their vulnerability to cumulative exposure. These results are consistent with prior research linking low literacy to suboptimal engagement in environmental and health-protective behaviors [39,36]. The association between reliance on bottled or filtered water and poorer health literacy may be partly explained by perceived protection. Individuals who depend on commercially treated or filtered water may assume that potential contaminants, including microplastics, are adequately removed, thereby reducing motivation to seek further information or develop broader risk awareness.

This interpretation is closely intertwined with socioeconomic and urban–rural contexts. Higher-income households, often residing in urbanized areas, are more likely to afford bottled or filtered water and to rely on technological solutions rather than personal preventive knowledge. Conversely, individuals using river or tap water may perceive higher exposure risk, which can stimulate more active engagement in health information seeking and preventive decision-making. These interconnected contextual factors may collectively shape health literacy related to microplastic contamination and should be interpreted cautiously.

The findings also highlight the limitations of relying solely on technological interventions to mitigate microplastic exposure. While advanced water treatment processes such as coagulation–flocculation and membrane filtration can remove a substantial fraction of microplastics, their efficiency declines markedly for smaller particles and varies by polymer type [24–28]. Empirical evidence from Thailand has confirmed the persistence of microplastics in both treated water supplies and commonly consumed freshwater organisms [30–32]. In this context, health literacy functions as a critical upstream determinant, enabling individuals and communities to adopt preventive strategies that complement infrastructural controls and reduce reliance on downstream remediation alone.

From a policy and public health perspective, these findings underscore the necessity of integrating environmental health literacy into microplastic risk governance. Educational interventions that merely disseminate information are unlikely to yield sustained behavioral change unless they explicitly target higher-order competencies such as critical appraisal and decision-making. Community-based programs that leverage local knowledge, health volunteers, and culturally resonant communication strategies may be particularly effective in strengthening literacy and fostering informed environmental stewardship [14,39]. Aligning such initiatives with national public health and environmental policies could further support progress toward Sustainable Development Goals related to clean water and responsible consumption.

This study is strengthened by its large, regionally representative sample and rigorous multistage sampling across three major river basins, enhancing the external validity of the findings. The use of a theoretically grounded health literacy framework and psychometrically robust instruments further reinforces the reliability and interpretability of the results. Importantly, this research contributes novel evidence by empirically linking health literacy with microplastic-related preventive behaviors in a real-world community setting, addressing a critical gap in the environmental health literature.

Although occupational category, household size, and water source were considered as contextual indicators, this study did not aim to quantify occupation-specific or activity-based microplastic exposure. The primary focus was to examine how community-level health literacy shapes preventive behaviors. Future studies incorporating detailed occupational and exposure-specific subgroup analyses may provide further insights into differential microplastic risks.

Nevertheless, several limitations merit consideration. First, the cross-sectional design precludes causal inference and limits the ability to disentangle bidirectional relationships between health literacy and behavior. Second, this study focused on permanent residents to reflect stable community-level practices. However, transient and migrant populations with potentially different exposure pathways were not included, which may limit generalizability. Future research should incorporate mobile populations to better capture diverse microplastic exposure contexts. Third, the microplastic health literacy instrument used in this study represents an initial application. Although internal consistency reliability was acceptable, comprehensive construct validation, including factor analysis, was not performed. Therefore, domain-specific interpretations of health literacy should be made with caution, and further psychometric validation is warranted in future

studies. Forth, data collection during the dry season may also constrain generalizability, as seasonal hydrological dynamics can influence waste dispersion, water use patterns, and risk perception [12]. Last, reliance on self-reported behaviors introduces the potential for social desirability bias, which may partially explain the relatively high prevalence of reported preventive practices despite widespread insufficient literacy. Future research employing longitudinal designs, seasonal comparisons, and objective exposure assessments would provide a more nuanced understanding of how health literacy shapes microplastic exposure and prevention over time.

## Conclusion and recommendations

This study highlights health literacy as a key determinant of preventive behaviors against microplastic contamination among riverine communities in Northeastern Thailand. Despite the presence of reported preventive practices, nearly half of the participants demonstrated insufficient health literacy, particularly in cognitive understanding and decision-making. This gap suggests that preventive behaviors are not consistently grounded in informed and sustainable decision-making.

Marked disparities in health literacy and preventive behaviors were observed across community types and demographic characteristics, with rural and semi-urban residents being disproportionately affected. Insufficient health literacy emerged as the strongest predictor of poor preventive behaviors, underscoring its central role in shaping individual and community responses to environmental exposure. These findings indicate that technological and infrastructural interventions alone are insufficient to reduce microplastic-related health risks without concurrent improvements in population health literacy.

Accordingly, environmental health literacy should be integrated into public health and environmental management strategies, with targeted, community-based interventions focusing on critical appraisal and decision-making skills. Strengthening environmental education across the life course and aligning literacy-focused interventions with water and waste management policies are essential for sustainable microplastic exposure prevention. Future research should employ longitudinal and mixed-methods approaches to further elucidate causal pathways and inform evidence-based policy development.

## Acknowledgments

The authors would like to express their sincere gratitude to Mahasarakham University, Thailand and Uppsala University, Sweden for support throughout this study. Special thanks are offered to all the community leaders, public health volunteers, public health students, and local residents living along the Chi, Mun, and Mekong River basins, whose cooperation and participation made this study possible. The authors also acknowledge the valuable assistance of the research assistants and enumerators who contributed to data collection and verification. Finally, the authors wish to thank their colleagues and peer reviewers for their constructive feedback, which greatly improved the quality of this manuscript.

## Author contributions

**Conceptualization:** Santisith Khiewkhern, Nitikorn Phoosuwan.

**Data curation:** Santisith Khiewkhern, Supatra Noo-In, Ruchakron Kongmant.

**Formal analysis:** Santisith Khiewkhern, Supatra Noo-In.

**Funding acquisition:** Santisith Khiewkhern, Nitikorn Phoosuwan.

**Investigation:** Supatra Noo-In, Chitkamon Srichomphoo, Jirarat Ruetrakul.

**Methodology:** Santisith Khiewkhern, Jirarat Ruetrakul, Nitikorn Phoosuwan.

**Project administration:** Santisith Khiewkhern, Nitikorn Phoosuwan.

**Resources:** Santisith Khiewkhern.

**Software:** Santisith Khiewkhern.

**Supervision:** Santisith Khiewkhern, Nitikorn Phoosuwan.

**Validation:** Santisith Khiewkhern.

**Visualization:** Nitikorn Phoosuwan.

**Writing – original draft:** Santisith Khiewkhern, Chitkamon Srichomphoo, Ruchakron Kongmant.

**Writing – review & editing:** Jirarat Ruetrakul, Nitikorn Phoosuwan.

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
