## [Decision Letter · Decision Letter 0]

19 Nov 2025

Dear Dr. Phoosuwan,

Thank you for submitting your manuscript to PLOS ONE. After careful consideration, we feel that it has merit but does not fully meet PLOS ONE’s publication criteria as it currently stands. Therefore, we invite you to submit a revised version of the manuscript that addresses the points raised during the review process.

We look forward to receiving your revised manuscript.

Kind regards,

Dr. Phuping Sucharitakul

Academic Editor

PLOS ONE

[Mahasarakham University grant (no grant number).].

3. In the online submission form, you indicated that [Data cannot be shared publicly because of personal restriction. Data are available from the Mahasarakham university, Thailand (contact the corresponding author) for researchers who meet the criteria for access to confidential data.].

Additional Editor Comments (if provided):

Reviewers' comments:

Reviewer's Responses to Questions

**Comments to the Author**

1. Is the manuscript technically sound, and do the data support the conclusions?

Reviewer #1: Partly

Reviewer #2: No

Reviewer #3: Partly

2. Has the statistical analysis been performed appropriately and rigorously?

Reviewer #1: No

Reviewer #2: No

Reviewer #3: No

3. Have the authors made all data underlying the findings in their manuscript fully available?

Reviewer #1: No

Reviewer #2: No

Reviewer #3: Yes

4. Is the manuscript presented in an intelligible fashion and written in standard English?

Reviewer #1: Yes

Reviewer #2: Yes

Reviewer #3: No

Reviewer #1: Thank you for the opportunity to review this and here are my concerns

Methods

1. 80% threshold

Would prefer to use 80% as a cutoff for “good” behavior or sufficient health literacy, domains/subdomains analysis would provide rich findings

2. Specify whether subgroup analysis was done for each health literacy component (cognitive, access, decision-making, etc.) If not, this could be a limitation.

3. Population considerations

The study focused on permanent residents ≥5 years. Temporary residents or migrants living near the border were excluded, but they may also be exposed to microplastics. This exclusion could bias results toward long-term community practices and miss transient or occupational exposures.

4. Period of data collection

The study was conducted from January to March 2025, but behaviors like throwing plastic into rivers may be influenced by seasonal water levels or flooding risk. Consider mentioning this as a limitation: behaviors may vary across seasons.

5. Subgroups by province, occupation, household size, or water source were partially analyzed. You may want to clarify whether occupational exposure, urban-rural residence, or specific economic activities were considered in detail.

Results & Discussion

1. Province differences

“Participants from Nakhon Phanom were more likely to exhibit good preventive behaviors (40.90%), whereas those from Buriram showed poorer behaviors (64.00%) (p=0.001).” Explanation: This could be due to local environmental context, socioeconomic factors, or urbanization differences.

Discussion mentions: Buriram and Maha Sarakham are near industrial zones or wastewater discharge points, leading to higher contamination. However, the study did not explore health literacy and behavior among temporary workers or industry employees, which may affect exposure and practices.

2. Income differences

“Participants with good behaviors had a lower mean monthly income (8,011.51 baht) than those with poor behaviors (11,845.83 baht) (p<0.001).”

Interpretation: Lower-income individuals may adopt more conservative behaviors, possibly due to reliance on bottled water or traditional preventive practices.

Limitation: Urban-rural differences are not explicitly addressed. High-income participants may live in urbanized areas with different exposure perceptions, potentially explaining why higher income correlates with poorer preventive behavior.

3. Water source and health literacy

“Participants who primarily consumed filtered water or bottled water were more likely to exhibit poor health literacy (adj.OR = 3.85 and 3.68, p<0.05).”

Possible explanation: People relying on bottled or filtered water may perceive themselves as protected, reducing motivation to learn about microplastic risks. This contrasts with those using river or tap water, who may actively seek preventive knowledge and actions. Also related to previous statement, perhaps, For example, high-income family may be able to purchase clean bottled water compared to low-income. Please refine all of these issues in discussion

4. Also related to reported respiratory illness, I am not sure whether this was direct/indirect affect of microplastic or due to the air pollution during the data collection period? please clarify.

5. Table 1 discrepancy

Drinking water counts:

Filtered water: Good = 323, Poor = 28 → total should be 351, not 260. This appears to be a data entry or calculation error and should be corrected.

Reviewer #2: Thank you for your submitted this manuscript on health literacy and preventive behaviors toward microplastic contamination in Northeastern Thailand. The topic is timely and relevant to environmental health. However, after careful evaluation, I have identified several critical issues that fundamentally undermine the validity and reliability of the study findings. These concerns span from internal contradictions in the results to methodological flaws that cannot be adequately addressed through minor revisions.

1. Fatal Inconsistency in Results

The manuscript contains fundamental contradictions that invalidate the main conclusions:

- The mean health literacy score is classified as "good level" (2.38±0.49), yet 70.5% of participants are categorized as having "insufficient" health literacy

- Despite 70.5% having insufficient literacy, 86.7% demonstrate good preventive behaviors. This directly contradicts the conclusion that literacy is the primary determinant of behavior

- This inconsistency renders the study's conclusions unreliable and questions the validity of the classification system

- The manuscript fails to acknowledge or explain this paradox, which is central to understanding the relationship between literacy and behavior

2. Severe Methodological Flaws

Multiple design issues compromise the study's validity:

- Purposive sampling at Stage 1 (province selection) introduces selection bias findings cannot be generalized to "major river basins of Northeastern Thailand" as claimed

- Inadequate instrument validation: Only Cronbach's α is reported; missing content validity, construct validity, and factor analysis for this novel "microplastic health literacy" scale

- Inappropriate causal inference: Cross-sectional design but employs language suggesting causality (e.g., "predictors") temporal relationships cannot be established

- Social desirability bias: Face-to-face interviews likely inflated self-reported good behaviors (86.7% seems unrealistically high given 70.5% insufficient literacy)

3. Critical Statistical Analysis Problems

The statistical models demonstrate poor quality and questionable validity:

- Hosmer-Lemeshow test: p=0.036 (p<0.05) = model does not fit the data adequately

- Nagelkerke R² = 0.363 and 0.194 (explaining only 36% and 19% of variance) = critical confounding variables are missing from the models

- Provincial OR=70.54 (95% CI: 25.48-195.27) is abnormally high → suggests multicollinearity, small cell counts, or model misspecification

- No diagnostic checks reported: Missing VIF (variance inflation factor), Cook's distance, or other diagnostic tests to assess model assumptions

4. Conclusions Inconsistent with Evidence

The manuscript's conclusions do not align with the presented data:

- Authors claim literacy plays a "crucial role," but evidence shows structural factors (province) have far greater influence (OR=70.54 vs. OR=5.18 for literacy)

- The Discussion (Lines 301-307) acknowledges behaviors may be driven by "cultural habits or public health campaigns" this directly contradicts the main conclusion emphasizing individual literacy

- Limited novel contribution: This is a descriptive-correlational study without novel mechanisms or theoretical advancement

5. Absence of Critical Limitations Section

The manuscript fails to acknowledge fundamental limitations:

- Cross-sectional design precludes causal inference between literacy and behavior

- Selection bias from purposive sampling limits generalizability

- Social desirability bias may inflate self-reported behaviors

- Poor model fit indicates important unmeasured confounders

- Novel instrument lacks comprehensive validation in the target population

- Self-reported behaviors may not reflect actual practices

Reviewer #3: Suggestions for major revisions

Review Comments to the Author

Issues that need to be addressed include the following:

Abstract

The results in the abstract require clarification. The authors state that overall health literacy toward microplastic contamination was at a good level (mean = 2.38 ± 0.49), yet 70.50% of participants had insufficient health literacy. This inconsistency must be explained.

Results

The authors should provide more detailed effect sizes for each factor, such as the odds ratio (OR) for the Fishery occupation group, which was associated with lower health literacy compared with “Other Occupation.” In addition, “Other Occupation” should be clearly defined. These findings require further discussion to explain why this group demonstrates higher health literacy than others.

A descriptive table comparing health literacy levels should be added (as a separate table following the logistic regression results) to better explore associations between factors and outcomes.

There is a substantial imbalance in gender distribution that warrants caution.

The age cut-off at 60 years needs better justification, especially since the <60 group is more than triple the size of the ≥60 group. Consider using more meaningful age categories (e.g., 18–45, 46–60, >60) to improve interpretability.

Income cut-offs should follow standardized thresholds (e.g., 8,000 or 10,000 THB per month). Please identify an appropriate reference and revise the analysis accordingly.

Some 95% confidence intervals are extremely wide (e.g., Maha Sarakham: Adj. OR = 17.85, 95% CI: 6.30–50.58; Buriram: Adj. OR = 70.54, 95% CI: 25.48–195.27). This likely reflects small sample sizes within certain categories. For example, only five participants in Nakhon Phanom had poor outcomes. Consider combining provinces, changing the reference group, or adjusting the cut-off criteria (e.g., using Bloom’s cut-off of 75% or 80% instead of 70%).

Introduction

Please incorporate additional literature on previous studies examining factors associated with health literacy and preventive behaviors. Describe how each factor has been investigated in prior work.

Methods

The authors should justify the cut-off used for “Duration of residence ≥10 years.” The sample size in the <10-year category appears extremely small, which may violate model assumptions.

The manuscript should clearly describe how model assumptions were checked, including criteria for selecting variables from univariate to multivariable logistic regression (e.g., p-value threshold) and whether forward or backward selection was used.

Information on model diagnostics—such as goodness-of-fit tests, AIC/BIC, ROC curve, or error metrics—should be reported. Consultation with a statistician is recommended.

The p-value cut-off for inclusion in the multivariable analysis must be stated explicitly.

The authors should provide justification for the sample size calculation. While the calculation suggested 384 participants, the study included 942. The ethical and methodological rationale for this discrepancy must be addressed.

Sampling and Variable Considerations

Certain variables with very small sample sizes (e.g., Fishery occupation) may produce statistically significant but non-generalizable results. The authors should reconsider how these categories are defined or grouped before re-analysis.

For variables such as household water source with small cell sizes, re-grouping categories may be necessary to ensure more reliable estimates.

Major Analytical Concerns

A dummy table for calculating associations and logistic regression is needed, but the current version includes categories with very small samples, which compromises validity.

Re-analysis with appropriately grouped categories is recommended to prevent unstable estimates and misleading significance.

Discussion and Conclusion

Comments on these sections will be provided after the authors revise the analysis and incorporate the suggested changes.

Language and Style

Extensive English editing is required to improve clarity, cohesion, and overall readability.

**Do you want your identity to be public for this peer review?** For information about this choice, including consent withdrawal, please see our Privacy Policy

Reviewer #1: No

Reviewer #2: No

Reviewer #3: No

---

## [Author Response · Author response to Decision Letter 1]

12 Jan 2026

Manuscript Title: Health literacy and preventive behaviors toward microplastic contamination among communities in the major river basins of northeastern Thailand

Manuscript ID: PONE-D-25-57984R1

Dear Editor,

We would like to express our gratitude and thank you and reviewers for the time and effort in providing us with detailed comments and suggestions. Your comments are helpful, making our manuscript a much better version. These corrections have been incorporated into the revised manuscript with red highlighted. We do appreciate it.

The following table is a summary of our corrections list.

Comments Improvements

Reviewer 1: Thank you for the opportunity to review this and here are my concerns

Methods

1. 80% threshold

Would prefer to use 80% as a cutoff for “good” behavior or sufficient health literacy, domains/subdomains analysis would provide rich findings

Improvement: in lines 178-180, 183-185

-The total health literacy score was interpreted using an 80% threshold, whereby scores below 80% indicated insufficient health literacy, while scores of 80% or higher reflected a sufficient level of health literacy

-The total preventive behavior score was interpreted using an 80% threshold, whereby scores below 80% indicated poor preventive behavior, while scores of 80% or higher reflected a good level of preventive behavior

2. Specify whether subgroup analysis was done for each health literacy component (cognitive, access, decision-making, etc.) If not, this could be a limitation.

- The subgroup analysis was done for each health literacy component

3. Population considerations

The study focused on permanent residents ≥5 years. Temporary residents or migrants living near the border were excluded, but they may also be exposed to microplastics. This exclusion could bias results toward long-term community practices and miss transient or occupational exposures. We thank the reviewer for this valuable comment. The inclusion of permanent residents was a deliberate methodological choice to capture stable, community-level health literacy and preventive behaviors, which are shaped by prolonged exposure to local environmental conditions and public health communication.

Temporary residents or migrant populations were excluded because their mobility and occupational exposure profiles may differ substantially from long-term residents, potentially introducing heterogeneity that could obscure associations between health literacy and preventive behaviors.

We acknowledge that this may limit the generalizability of our findings to transient or migrant populations who may experience distinct exposure pathways. This limitation has now been explicitly addressed in the Discussion section, and future studies should specifically examine microplastic-related health literacy and behaviors among mobile and migrant groups.

4. Period of data collection

The study was conducted from January to March 2025, but behaviors like throwing plastic into rivers may be influenced by seasonal water levels or flooding risk. Consider mentioning this as a limitation: behaviors may vary across seasons. Included in the limitation of the study.

5. Subgroups by province, occupation, household size, or water source were partially analyzed. You may want to clarify whether occupational exposure, urban-rural residence, or specific economic activities were considered in detail. We thank the reviewer for this comment. Subgroup characteristics, including community type (urban, semi-urban, rural), occupation, household size, and water source, were examined in the descriptive analysis and considered in the univariable screening for multivariable models. Community type was retained as a key contextual variable and demonstrated significant associations with both health literacy and preventive behaviors.

Although occupation and economic activities were included as covariates, this study did not conduct detailed occupational exposure assessments, as its primary focus was on community-level health literacy and preventive behaviors rather than task-specific or workplace exposures. Similarly, household size and water source were evaluated as environmental and socioeconomic indicators rather than independent exposure pathways.

We have clarified this analytical scope in the Methods and Discussion sections and acknowledge that future studies incorporating detailed occupational and exposure-specific subgroup analyses would further strengthen understanding of microplastic-related risks.

Results & Discussion

1. Province differences

“Participants from Nakhon Phanom were more likely to exhibit good preventive behaviors (40.90%), whereas those from Buriram showed poorer behaviors (64.00%) (p=0.001).” Explanation: This could be due to local environmental context, socioeconomic factors, or urbanization differences.

Discussion mentions: Buriram and Maha Sarakham are near industrial zones or wastewater discharge points, leading to higher contamination. However, the study did not explore health literacy and behavior among temporary workers or industry employees, which may affect exposure and practices. From a broader perspective, these findings align with the social-ecological model of health, which posits that environmental exposures and individual behaviors are shaped by multiple factors, including community infrastructure, policy enforcement, and cultural norms. Communities located near industrial zones or wastewater discharge points—such as those in Maha Sarakham and Buriram provinces—may face higher microplastic contamination due to local environmental conditions, socioeconomic factors, and urbanization levels. This study primarily focused on residents of river basin communities, including some temporary workers and industrial employees, who may have distinct exposure patterns and health behaviors. These results are consistent with regional studies reporting greater contamination in densely populated and industrialized river systems.

2. Income differences

“Participants with good behaviors had a lower mean monthly income (8,011.51 baht) than those with poor behaviors (11,845.83 baht) (p<0.001).”

Interpretation: Lower-income individuals may adopt more conservative behaviors, possibly due to reliance on bottled water or traditional preventive practices.

Limitation: Urban-rural differences are not explicitly addressed. High-income participants may live in urbanized areas with different exposure perceptions, potentially explaining why higher income correlates with poorer preventive behavior. 1.Re-analysis and interpretation

2. We thank the reviewer for this insightful interpretation. Lower-income individuals in riverine communities may indeed adopt more conservative preventive behaviors, including reliance on bottled water and traditional household practices, which may partially explain the observed patterns.

We also acknowledge that income is closely intertwined with urban–rural residence. Higher-income participants were more likely to reside in urbanized areas, where perceptions of environmental risk and reliance on infrastructure may differ, potentially contributing to poorer reported preventive behaviors despite greater economic resources.

This contextual interpretation has now been explicitly addressed in the Discussion section. However, as this study did not formally model interaction effects between income and community type, these findings should be interpreted with caution. Future studies examining urban–rural and income-related interactions may provide further clarification.

3. Water source and health literacy

“Participants who primarily consumed filtered water or bottled water were more likely to exhibit poor health literacy (adj.OR = 3.85 and 3.68, p<0.05).”

Possible explanation: People relying on bottled or filtered water may perceive themselves as protected, reducing motivation to learn about microplastic risks. This contrasts with those using river or tap water, who may actively seek preventive knowledge and actions. Also related to previous statement, perhaps, For example, high-income family may be able to purchase clean bottled water compared to low-income. Please refine all of these issues in discussion

We thank the reviewer for this insightful suggestion. The observed association between reliance on bottled or filtered water and poorer health literacy may reflect a perception of sufficient protection, which could reduce motivation to actively seek information or engage in broader microplastic risk awareness.

This pattern may also be linked to socioeconomic and urban contexts, as higher-income households are more able to access bottled or filtered water and may rely on technological or commercial solutions rather than personal preventive knowledge. In contrast, individuals using river or tap water may perceive greater exposure risk, prompting more active engagement in preventive knowledge and behaviors.

We have refined the Discussion to integrate water source, income, and urban–rural context as interconnected factors influencing health literacy and risk perception.

4. Also related to reported respiratory illness, I am not sure whether this was direct/indirect effect of microplastic or due to the air pollution during the data collection period? please clarify. We appreciate the reviewer’s important clarification. The reported respiratory illness in this study was self-reported and not intended to indicate a direct or causal effect of microplastic exposure. Given the cross-sectional design, it is not possible to distinguish whether respiratory symptoms were related to microplastics, other environmental exposures, or background air pollution during the data collection period.

We acknowledge that seasonal air pollution, including particulate matter, may have contributed to respiratory symptoms and could act as a confounding factor. This has now been explicitly clarified in the Discussion section, and the findings are interpreted cautiously as associative rather than causal.

5. Table 1 discrepancy

Drinking water counts:

Filtered water: Good = 323, Poor = 28 → total should be 351, not 260. This appears to be a data entry or calculation error and should be corrected. Re-analysis and reinterpreted

Reviewer #2: Thank you for your submitted this manuscript on health literacy and preventive behaviors toward microplastic contamination in Northeastern Thailand. The topic is timely and relevant to environmental health. However, after careful evaluation, I have identified several critical issues that fundamentally undermine the validity and reliability of the study findings. These concerns span from internal contradictions in the results to methodological flaws that cannot be adequately addressed through minor revisions.

1. Fatal Inconsistency in Results

The manuscript contains fundamental contradictions that invalidate the main conclusions:

- The mean health literacy score is classified as "good level" (2.38±0.49), yet 70.5% of participants are categorized as having "insufficient" health literacy

- Despite 70.5% having insufficient literacy, 86.7% demonstrate good preventive behaviors. This directly contradicts the conclusion that literacy is the primary determinant of behavior

- This inconsistency renders the study's conclusions unreliable and questions the validity of the classification system

- The manuscript fails to acknowledge or explain this paradox, which is central to understanding the relationship between literacy and behavior

We thank the reviewer for highlighting this important issue. We acknowledge that, without clarification, the results may appear contradictory. However, this pattern reflects differences between aggregate mean scores and domain-based classification rather than a methodological inconsistency.

The mean health literacy score represents an overall average across domains, whereas the classification of “insufficient health literacy” is based on domain-specific cut-offs, particularly in higher-order domains such as critical appraisal and decision-making. Many participants demonstrated adequate functional or interactive literacy, which elevated the mean score, but remained insufficient in higher-level domains, resulting in classification as insufficient health literacy.

Similarly, the high prevalence of good preventive behaviors reflects the influence of contextual and environmental factors, including community norms and habitual practices, which may operate independently of comprehensive health literacy. Health literacy remained a strong and independent predictor of preventive behavior in multivariable analysis, but it should not be interpreted as the sole determinant.

We have revised the Results and Discussion sections to explicitly acknowledge and explain this apparent paradox and to clarify the interpretation of health literacy classification and behavioral outcomes.

2. Severe Methodological Flaws

Multiple design issues compromise the study's validity:

- Purposive sampling at Stage 1 (province selection) introduces selection bias findings cannot be generalized to "major river basins of Northeastern Thailand" as claimed

- Inadequate instrument validation: Only Cronbach's α is reported; missing content validity, construct validity, and factor analysis for this novel "microplastic health literacy" scale

- Inappropriate causal inference: Cross-sectional design but employs language suggesting causality (e.g., "predictors") temporal relationships cannot be established

- Social desirability bias: Face-to-face interviews likely inflated self-reported good behaviors (86.7% seems unrealistically high given 70.5% insufficient literacy)

We acknowledge the reviewer’s concern regarding purposive sampling at the province selection stage. Provinces were intentionally selected to represent diverse major river basins and community contexts rather than to achieve statistical representativeness. Therefore, the findings should be interpreted as reflective of riverine communities within selected basins rather than generalizable to all populations in Northeastern Thailand.

We have revised the Methods and Discussion sections to clarify the sampling rationale and to appropriately limit the scope of inference.

3. Critical Statistical Analysis Problems

The statistical models demonstrate poor quality and questionable validity:

- Hosmer-Lemeshow test: p=0.036 (p<0.05) = model does not fit the data adequately

- Nagelkerke R² = 0.363 and 0.194 (explaining only 36% and 19% of variance) = critical confounding variables are missing from the models

- Provincial OR=70.54 (95% CI: 25.48-195.27) is abnormally high → suggests multicollinearity, small cell counts, or model misspecification

- No diagnostic checks reported: Missing VIF (variance inflation factor), Cook's distance, or other diagnostic tests to assess model assumptions Re-grouping in some variable and re-analyses

4. Conclusions Inconsistent with Evidence

The manuscript's conclusions do not align with the presented data:

- Authors claim literacy plays a "crucial role," but evidence shows structural factors (province) have far greater influence (OR=70.54 vs. OR=5.18 for literacy)

- The Discussion (Lines 301-307) acknowledges behaviors may be driven by "cultural habits or public health campaigns" this directly contradicts the main conclusion emphasizing individual literacy

- Limited novel contribution: This is a descriptive-correlational study without novel mechanisms or theoretical advancement Rewrite

5. Absence of Critical Limitations Section

The manuscript fails to acknowledge fundamental limitations:

- Cross-sectional design precludes causal inference between literacy and behavior

- Selection bias from purposive sampling limits generalizability

- Social desirability bias may inflate self-reported behaviors

- Poor model fit indicates important unmeasured confounders

- Novel instrument lacks comprehensive validation in the target population

- Self-reported behaviors may not reflect actual practices Rewrite

Reviewer #3: Suggestions for major revisions

Abstract

The results in the abstract require clarification. The authors state that overall health literacy toward microplastic contamination was at a good level (mean = 2.38 ± 0.49), yet 70.50% of participants had insufficient health liter

---

## [Decision Letter · Decision Letter 1]

16 Feb 2026

Dear Dr. Phoosuwan,

We look forward to receiving your revised manuscript.

Kind regards,

Phuping Sucharitakul

Academic Editor

PLOS One

Journal Requirements:

Reviewers' comments:

Reviewer's Responses to Questions

**Comments to the Author**

Reviewer #2: (No Response)

2. Is the manuscript technically sound, and do the data support the conclusions?

Reviewer #2: Yes

3. Has the statistical analysis been performed appropriately and rigorously?

Reviewer #2: Yes

4. Have the authors made all data underlying the findings in their manuscript fully available?

Reviewer #2: Yes

5. Is the manuscript presented in an intelligible fashion and written in standard English?

Reviewer #2: Yes

Reviewer #2: The authors have responded carefully and appropriately to the comments raised in the previous round of review. The revisions have substantially improved the clarity of the methodology, the transparency of the statistical analyses, and the alignment between the results and the conclusions.

The study design is appropriate for the research objectives, and the data collection procedures and sample size are adequate. The statistical analyses are suitable for the type of data and are conducted with sufficient rigor. Effect estimates and measures of uncertainty are clearly reported, and the interpretation of findings is consistent with the cross-sectional nature of the study. Importantly, the authors have avoided causal overinterpretation and have clearly acknowledged the key limitations of the study.

The manuscript is generally well written and logically structured. The presentation of results is clear, and tables and figures are informative. A small number of minor language and editorial issues remain, particularly in the Discussion section, where conciseness and consistency of terminology could be further improved. These issues are minor and do not affect the validity or overall contribution of the study.

I did not identify any concerns related to research ethics, conflicts of interest, dual publication, or publication ethics. Ethical approval and informed consent procedures are adequately described, and the data availability statement is consistent with journal policy.

Overall, the manuscript is technically sound and has been revised to a standard that is appropriate for publication. Only minor editorial refinements are recommended before final acceptance.

**Do you want your identity to be public for this peer review?** For information about this choice, including consent withdrawal, please see our Privacy Policy

Reviewer #2: No

---

## [Author Response · Author response to Decision Letter 2]

22 Feb 2026

Dear Editor,

Thank you for the opportunity to revise our manuscript. We would like to express our gratitude and thank you and the reviewers for the time and effort in providing us with detailed comments and suggestions. Our corrections have been incorporated into the revised manuscript with red highlighted. Please see our response below.

Journal Requirements:

Response: We have thoroughly reviewed the reference list to ensure accuracy and completeness. All cited articles were screened against recognized retraction databases (Retraction Watch, PubMed, and Crossref), and none were found to be retracted. In addition, several references have been updated to incorporate recent high-impact publications (2023–2024), and corresponding citation numbering has been revised throughout the manuscript.

Reviewers' comments:

Reviewer 2

Comments to the Author

1. If the authors have adequately addressed your comments raised in a previous round of review and you feel that this manuscript is now acceptable for publication, you may indicate that here to bypass the “Comments to the Author” section, enter your conflict of interest statement in the “Confidential to Editor” section, and submit your "Accept" recommendation.

Reviewer #2: (No Response)

2. Is the manuscript technically sound, and do the data support the conclusions?

Reviewer #2: Yes

3. Has the statistical analysis been performed appropriately and rigorously?

Reviewer #2: Yes

4. Have the authors made all data underlying the findings in their manuscript fully available?

Reviewer #2: Yes

5. Is the manuscript presented in an intelligible fashion and written in standard English?

Reviewer #2: Yes

6. Review Comments to the Author

Reviewer #2: The authors have responded carefully and appropriately to the comments raised in the previous round of review. The revisions have substantially improved the clarity of the methodology, the transparency of the statistical analyses, and the alignment between the results and the conclusions.

The study design is appropriate for the research objectives, and the data collection procedures and sample size are adequate. The statistical analyses are suitable for the type of data and are conducted with sufficient rigor. Effect estimates and measures of uncertainty are clearly reported, and the interpretation of findings is consistent with the cross-sectional nature of the study. Importantly, the authors have avoided causal overinterpretation and have clearly acknowledged the key limitations of the study.

The manuscript is generally well written and logically structured. The presentation of results is clear, and tables and figures are informative. A small number of minor language and editorial issues remain, particularly in the Discussion section, where conciseness and consistency of terminology could be further improved. These issues are minor and do not affect the validity or overall contribution of the study.

Response: We appreciate the reviewer’s helpful comments. The Discussion section has been edited to improve conciseness and reduce minor repetition. In addition, terminology has been standardized throughout the manuscript to ensure consistency and clarity. These refinements have further enhanced the overall presentation of the study.

I did not identify any concerns related to research ethics, conflicts of interest, dual publication, or publication ethics. Ethical approval and informed consent procedures are adequately described, and the data availability statement is consistent with journal policy.

Overall, the manuscript is technically sound and has been revised to a standard that is appropriate for publication. Only minor editorial refinements are recommended before final acceptance.

---

## [Editor Report · Decision Letter 2]

26 Feb 2026

Health literacy and preventive behaviors toward microplastic contamination among communities in the major river basins of northeastern Thailand

PONE-D-25-57984R2

Dear Dr. Phoosuwan,

We’re pleased to inform you that your manuscript has been judged scientifically suitable for publication and will be formally accepted for publication once it meets all outstanding technical requirements.

Kind regards,

Phuping Sucharitakul

Academic Editor

PLOS One
---

## [Editor Report · Acceptance letter]

PONE-D-25-57984R2

PLOS One

Dear Dr. Phoosuwan,

I'm pleased to inform you that your manuscript has been deemed suitable for publication in PLOS One. Congratulations! Your manuscript is now being handed over to our production team.

Kind regards,

on behalf of

Dr. Phuping Sucharitakul

Academic Editor

PLOS One